# Towards Free Data Selection with General-Purpose Models

**Yichen Xie, Mingyu Ding**[*]**, Masayoshi Tomizuka, Wei Zhan**
UC Berkeley
{yichen_xie, myding, tomizuka, wzhan}@berkeley.edu

## Abstract

A desirable data selection algorithm can efficiently choose the most informative samples to maximize the utility of limited annotation budgets. However, current approaches, represented by active learning methods, typically follow a cumbersome pipeline that iterates the time-consuming model training and batch data selection repeatedly. In this paper, we challenge this status quo by designing a distinct data selection pipeline that utilizes existing general-purpose models to select data from various datasets with a single-pass inference without the need for additional training or supervision. A novel free data selection (FreeSel) method is proposed following this new pipeline. Specifically, we define semantic patterns extracted from intermediate features of the general-purpose model to capture subtle local information in each image. We then enable the selection of all data samples in a single pass through distance-based sampling at the fine-grained semantic pattern level. FreeSel bypasses the heavy batch selection process, achieving a significant improvement in efficiency and being $530\times$ faster than existing active learning methods. Extensive experiments verify the effectiveness of FreeSel on various computer vision tasks. Our code is available at https://github.com/yichen928/FreeSel.

## 1 Introduction

Deep Neural Network (DNN) models have achieved remarkable progress in various tasks, benefiting from abundant training samples and labels. Unfortunately, data labeling tends to be time-consuming and costly, especially for dense prediction tasks such as object detection and semantic segmentation, where experts may spend up to 90 minutes per image [33]. As such, effectively exploiting the limited annotation budget has become a long-standing problem in the advancement of computer vision.

Many methods have been proposed to identify the most suitable samples for annotation, where the mainstream follows the active learning [43, 45] or subset selection [42] pipelines. However, both kinds of methods rely on task-specific models. As the most popular data selection strategy, active learning algorithms employ a time-consuming and computationally expensive batch selection strategy [44], as shown in Fig. 1a. Specifically, a task-specific model is first trained using a small initial set of labeled samples. Then, the model is utilized to select images within a specified batch budget size. These selected images are annotated and added to the labeled pool, after which the model is retrained or fine-tuned using all the labeled samples. This iterative process is repeated multiple times for a large unlabeled data pool. Since the selection of data is tightly coupled with the task-specific model, the entire pipeline needs to be restarted from scratch and repeated when working on different tasks or datasets. In many cases, it even requires up to *several days* to select sufficient samples from a medium-sized data pool (*e.g.*, Core-Set [44] in Tab. 1).

---

[*]Corresponding author.

37th Conference on Neural Information Processing Systems (NeurIPS 2023).

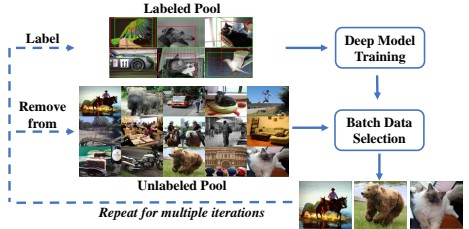
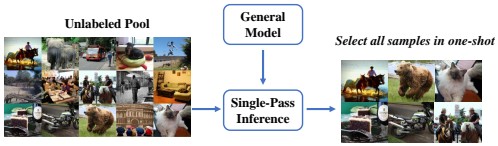

(a) **Active Learning Pipeline:** It follows the batch-selection strategy with iterative training.

(b) **Our Free Data Selection Pipeline:** Samples are selected in a single-pass without extra model training.

Figure 1: Comparisons between active learning pipeline and our proposed free selection pipeline.

In this paper, we challenge this *status quo* by introducing an efficient data selection pipeline that enables the selection of data within a single pass (as illustrated in Fig. 1b), therefore achieving comparable efficiency to random selection. We identify the efficiency bottleneck of data selection methods as the training of the task-specific model. Building upon insights from recent research on unsupervised learning [2, 64], we recognize that pretrained models [5, 63] possess the ability to encode the semantic information of images in a fine-grained level. This observation inspires us to integrate pretrained models into the data selection process, thereby decoupling data selection from task-specific models and leveraging the inherent diversity captured by pretrained models. By leveraging publicly available pretrained models, our pipeline incurs no additional training costs. To provide a concrete foundation for our design, we consider the following three guiding principles.

- **Generality:** We strive for decoupling data selection from task-specific models. It is desired that a *general* model works on the data selection of multiple tasks or datasets.
- **Efficiency:** The batch selection setting of active learning (Fig. 1a) is known to be time-consuming due to its iterative nature. It is expected to be replaced with a *single-pass* model inference on unlabeled data pools.
- **Non-supervision:** Annotators may not always respond in time, and the entire data selection progress may be delayed by frequent requests for labels. It is preferred that annotations are not required until the end of data selection.

In view of the above principles, we propose the *first* free data selection (FreeSel) method, to the best of our knowledge, satisfying all the above principles simultaneously. FreeSel selects data samples based on the diversity of local features. The features are extracted by a publicly available pretrained vision transformer [12], which is generic enough to facilitate data selection for different networks, datasets, and tasks after pretraining on large-scale datasets [11] in an unsupervised manner, *e.g.*, DINO [5]. We extract our newly defined semantic patterns by clustering the intermediate local features after an attention filter. The images are selected following a distance-based sampling strategy at the level of semantic patterns. In pursuit of efficiency, this whole process is finished within a single-pass model inference without any extra training. The data selection process is indeed unsupervised, which relieves the troubles of assigning responsive annotators.

As a result, our method pursues a *free* data selection using public pretrained models with a time efficiency close to random selection. We conduct extensive experiments on different tasks, datasets, and networks. When compared with existing active learning methods, our algorithm can achieve comparable performance with significantly advantageous efficiency.

Our contributions are three-fold. **1)** We for the first time, introduce a new free data selection pipeline that adheres to three important principles of *generality*, *efficiency*, and *non-supervision* with negligible time costs. **2)** We propose FreeSel, a novel method following our proposed pipeline. It can fill in the annotation budget in a single pass based on the inherent diversity of semantic patterns captured by pretrained models. **3)** Extensive experiments on image classification, object detection, and semantic segmentation demonstrate the effectiveness of our pipeline.

## 2 Related Work

**Active Learning.** Active learning aims to choose the most suitable samples for annotation so that model performance can be optimized with a limited annotation budget. Most existing work in this

Table 1: **Principles of Data Selection Methods:** *Task Model* refers to the coupling between data selection and a task-specific model. *Batch Selection* shows whether the method repeats the data selection in batch multiple times. *Multi-time Labeling* denotes whether it requests ground-truth labels in the data selection process. *Time* estimates the approximate time to select 8000 images from PASCAL VOC datasets (Sec. 5.5).

| Methods | Task Model | Batch Selection | Multi-time Labeling | Time |
|---|---|---|---|---|
| Core-Set [44] | ✓ | ✓ | ✓ | $\sim 42\ hours$ |
| Learn-Loss [57] | ✓ | ✓ | ✓ | + |
| CDAL [1] | ✓ | ✓ | ✓ | *label query* |
| FreeSel (ours) | ✗ | ✗ | ✗ | $285\ s$ ($\sim 530 \times$ faster) |

field [47, 57, 44, 18, 59, 60] follows a pool-based protocol, selecting samples based on the ranking of the whole dataset. There exist two mainstream sampling strategies for pool-based methods *i.e.* uncertainty and diversity. Uncertainty inside the model prediction reflects the difficulty of data samples, estimated by different heuristics such as probabilistic models [16, 13], entropy [26, 36], ensembles[3, 32], and loss function [57, 24]. Some other algorithms try to find the diverse subset which well represents the entire data pool. They measure the diversity with the Euclidean distance between global features [44], adversarial loss [47], or KL-divergence between local representations [1]. However, all these methods couple the data selection with a task model and require repetitive model training in the batch selection pipeline, resulting in inefficiency. Differently, our proposed pipeline selects samples through *a single-pass model inference* on each unlabeled pool.

**Subset Selection.** As another category of data selection algorithms, subset selection methods often select all the required samples in a single pass with the model trained on a labeled seed set. The subset is usually selected based on some criterion of uncertainty [27], diversity [6, 4], or their combination [42]. In contrast, our proposed pipeline needs neither extra training on the target dataset nor knowledge about the label space.

**Unsupervised Learning.** Both contrastive methods [17, 20, 61, 53, 25, 48, 39] and generative models [52, 19, 49] have achieved great success in unsupervised representation learning. Contrastive methods discriminate different images without using any explicit categories. In contrast, generative methods directly predict masked visual information inside images. We exploit a general pretrained model [5] to represent input images for task-agnostic data selection. As a result, we do not train models specific to each task like the traditional active learning pipeline.

**Data Selection with Pretrained Models.** There are some attempts to combine unsupervised pretraining and data selection. [56] selects data samples by the loss of pretext tasks, but requires different pretext tasks for different downstream tasks. [37] formulates active learning as an integer programming problem in the feature space, handling low-budget cases. [51] and [54] select samples based on the diversity of global features, targeted for semi-supervised learning and model finetuning settings respectively. Active labeling proposed in [23] is the most similar to our paper, but their method considers selective partial labeling in each sample instead of sample selection and is limited to 3D tasks with the same networks for pretraining and downstream tasks.

## 3 Preliminary Study: Off-the-Shelf Features for Data Selection

Active learning work [44, 1] often selects representative samples based on the features extracted by task-specific models trained separately for each task. A straightforward alternative is to use off-the-shelf features instead, which are extracted by general-purpose models pretrained on a large-scale dataset. If it performs well, we can trivially improve the efficiency by eliminating the training step on each dataset.

We conduct this preliminary study on the object detection task over the PASCAL VOC dataset [14].

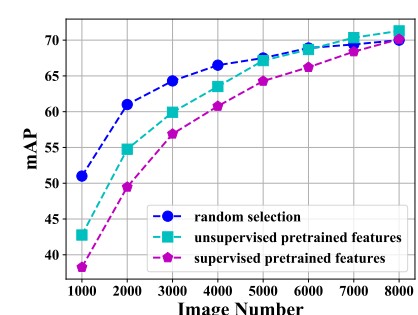

Figure 2: **Core-Set over Off-the-Shelf Features**

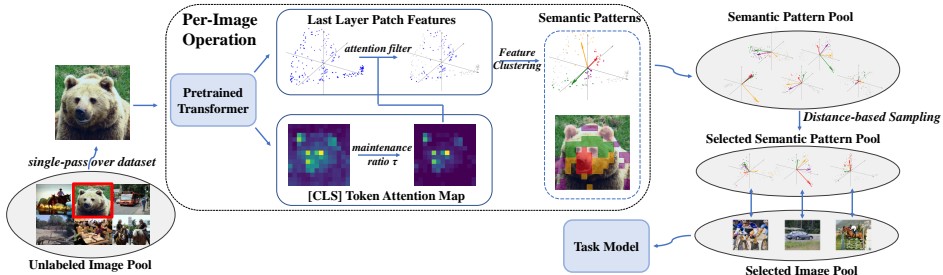

Figure 3: **Overview of Our Proposed FreeSel:** Our method uses a general pretrained vision transformer to extract features from images. Semantic patterns are derived from the intermediate features. Afterwards, we perform a distance-based sampling algorithm to select semantic patterns as well as the associated images. These selected images are labeled for downstream task model training.

Consistent with our following experiments, we
apply DeiT-S [2] [50] for feature extraction in data selection. The model is pretrained in either supervised or unsupervised (with DINO framework [5]) manners on ImageNet [11]. For data selection, we implement the classical Core-Set algorithm [44] over the extracted global features, *i.e.* the [CLS] token feature in the last layer. We use Core-Set with these features to select various numbers of training samples, and train object detection models (SSD-300 [34]) over the selected subsets.

Fig. 2 shows results in comparison with random selection. Unfortunately, we find that this naive combination of off-the-shelf features and Core-Set algorithms degrades the object detection performance, especially with relatively low sampling ratios. We consider two potential reasons for this failure: **1) Complex scenes are hard to represent globally.** Images may contain multiple objects including some very small ones. It is difficult for a global feature to represent all useful details in the image. **2) K-Center selects corner cases.** In the feature space, in order to cover all the data samples with a small radius, the K-Center algorithm of Core-Set tends to select all the corner cases.

The above two concerns motivate our design in Sec. 4. We represent each image with dense semantic patterns to maintain useful local information. Images are sampled based on some probability related to the distance between local semantic patterns to relieve the harmful preference for corner cases.

## 4 Methodology

We detail our new data selection method FreeSel, formulated in Sec. 4.1. We define a concept called *semantic pattern* in Sec. 4.2. Afterward, the sample selection strategy is explained in Sec. 4.3. An overview of our approach is illustrated in Fig. 3.

### 4.1 Formulation

We aim to select a diverse subset from the unlabeled data pool for annotation, which covers as much discriminative regional information in the original pool as possible. The regional information inside an image $I$ is reflected by the local features $\mathbf{f}^I = \{f_r^I | r = 1, 2, \ldots, HW\}$ of a pretrained DNN. $H, W$ are the height and width of the feature map. The $r$-th region feature $f_r^I \in \mathbb{R}^K$ in the feature map mainly describes the $r$-th region of the image [62, 41]. The discriminative power of all regional features $\mathbf{f}^I$ can be represented by countable knowledge points [31]. $f_r^I$ is considered as a knowledge point *w.r.t.* a pseudo-category $c$ if it is similar enough to the corresponding direction vector $\mu_c$.

$$p(c|f_r^I) = \frac{\pi_c \cdot p_{vMF}(f_r^I|c)}{\sum_{c'} \pi_{c'} \cdot p_{vMF}(f_r^I|c')} > \tau, \quad p_{vMF}(f|c) = C_d(\kappa_c) \cdot \exp(\kappa_c \cdot \cos(f_r^I, \mu_c)) \quad (1)$$

$c$ is a pseudo-category describing some specific visual patterns, *e.g.* an object part, which is represented by a vector $\mu_c$ in the feature space. $\pi_c$ is the prior probability of pseudo-category $c$, $\kappa_c$ is a concentration parameter, and $C_d(\kappa_c)$ is a normalizing constant.

Inversely, given knowledge points inside an image $I$, they can be clustered to estimate the $K$ pseudo-categories inside the image as $\hat{\mu}_j^I, j = 1, 2, \ldots, K$. We define the estimation as semantic patterns in

---

[2]We follow the name of networks in [50] in our paper. DeiT-S is also called ViT-small in [5].



Figure 4: **Visualization of Semantic Patterns:** Every two images are considered as a group. *Left:* The filtered local features (dots) of each image are grouped into semantic patterns (arrows). Gray features are eliminated in Eq. 2. Dimensions are reduced by PCA for visualization. *Right:* Regions inside images can be associated with corresponding local features and then semantic patterns.

---

**Algorithm 1: Semantic Pattern Extraction**

---

**Input:** Pretrained transformer $g$, unlabeled image pool $\mathcal{I}$, attention ratio $\tau$, centroid number $K$
**Output:** Semantic patterns $\hat{\boldsymbol{\mu}}^I = \{\hat{\mu}_j^I\}, I \in \mathcal{I}$

1 **for** $I \in \mathcal{I}$ **do**
2 $\quad$ $\mathbf{ca}^I, \mathbf{pa}^I, \mathbf{f}^I = g(I)$
$\quad$ /* last layer [CLS] and patch token attention and patch-wise features. $\quad$ */
3 $\quad$ Sort $ca_r^I, pa_r^I, f_r^I, r = 1, 2, \ldots, HW$ in the decreasing order of attention $ca_r^I$
4 $\quad$ $F(\mathbf{f}^I) = \{f_r^I | r = 1, 2, \ldots, t, \sum_{j=1}^{t} ca_j^I \leq \tau < \sum_{j=1}^{t+1} ca_j^I\}$
$\quad$ /* filter important regions based on the sorted attention (Eq. 2). $\quad$ */
5 $\quad$ Derive local similarity $\widehat{pa}_{ij}^I$ with Eq. 3
$\quad$ /* ignore attention between faraway regions. $\quad$ */
6 $\quad$ $\widehat{\mathbf{pa}}^I = [\widehat{pa}_{ij}^I]_{i,j=1,2,\ldots,t}$
$\quad$ /* only consider filtered $t$ regions. $\quad$ */
7 $\quad$ $\{C_j^I\}_{j=1}^{K} = SpectralCluster(\widehat{\mathbf{pa}}^I, K)$
$\quad$ /* divide $t$ regions into $K$ clusters with spectral clustering algorithm. $\quad$ */
8 $\quad$ $\hat{\mu}_j^I = \frac{1}{|C_j|} \sum_{r \in C_j} f_r^I, \qquad j = 1, 2, \ldots, K$
$\quad$ /* calculate the representation of each semantic pattern. $\quad$ */

---

Sec. 4.2. To ensure the diversity of our selection, our algorithm desires to find a subset of images $S_\mathcal{I}$ in Sec. 4.3, whose semantic patterns $\bigcup_{I \in S_\mathcal{I}} \{\hat{\mu}_j^I\}_{j=1}^{K}$ can be representative in the unlabeled pool.

### 4.2 Per-Image Semantic Patterns Extraction

To estimate the pseudo-categories, we define a novel notion called *semantic patterns*, which are extracted **from each image separately**. Given a pretrained vision transformer [12], we consider its last layer features for image $I$ as $\mathbf{f}^I = \{f_r^I\}_{r=1}^{HW}$, where each patch corresponds to a region $r$.

According to Eq. 1, only a few regional features may be considered as meaningful knowledge points, while other regions are useless or even distracting. However, it is non-trivial to distill these knowledge points without any information about the pseudo-categories. To this end, we resort to the [CLS] token self-attention map of the transformer, which serves as a natural filter for regional importance even without the supervision of category information [5].

**Attention Filter.** For image $I$, the last layer [CLS] token attention map (average of multi-heads) is denoted as $\mathbf{ca}^I = \{ca_r^I \in \mathbb{R}^+ | r = 1, 2, \ldots, HW\}, \sum_{r=1}^{HW} ca_r^I = 1$. We can filter the important regional features that jointly represent the most useful information in the entire image with Eq. 2.

$$F(\mathbf{f}^I) = \{f_r^I | r = 1, 2, \ldots, t, \sum_{j=1}^{t} ca_j^I \leq \tau < \sum_{j=1}^{t+1} ca_j^I\} \tag{2}$$

where regions $r = 1, 2, \ldots, HW$ are sorted in the **decreasing order** of $ca_r^I$, and $\tau \in (0, 1)$ is a hyper-parameter, meaning the **maintenance ratio** of information represented by the filtered important features. The filtered features $F(\mathbf{f}^I)$ are considered as the knowledge points inside the images.

**Feature Clustering.** To estimate the feature vectors for pseudo-categories, we perform clustering over the filtered $t$ knowledge points **inside each image separately**. Since K-Means is unreli-

able in the high-dimensional feature space (details in supplementary materials), we adopt spectral clustering instead. The self-attention map provides strong cues about the region-wise similarity inside each image. We denote the last layer attention map between patch tokens for image $I$ as $\mathbf{pa}^I = \left[pa_{ij}^I \in \mathbb{R}\right]_{i,j=1,2,\ldots,HW}, \sum_{j=1}^{HW} pa_{ij}^I = 1, \forall i$. It is more likely for nearby regions to interact with each other, so we only consider the attention between nearby patches [22].

$$\widehat{pa}_{ij}^I = \begin{cases} pa_{ij}^I & d(i,j) \leq d_0 \\ 0 & d(i,j) > d_0 \end{cases} \tag{3}$$

where $d(i,j)$ is the Chebyshev distance between regions $i, j$ in the feature map. We empirically set the threshold $d_0 = 2$ in our experiments. Besides, we only consider the $t$ regions after the filter in Eq. 2. In this case, we denote the new similarity matrix between patches as $\widehat{\mathbf{pa}}^I = \left[\widehat{pa}_{ij}^I\right]_{i,j=1,2,\ldots,t}$.

With this above $t \times t$ similarity matrix, we utilize spectral clustering algorithms [38, 55] to divide the remaining $t$ regions after filtering (Eq. 2) into $K$ clusters $C_j, j = 1, 2, \ldots, K$, each corresponding to a pseudo-category, where $K$ is a hyper-parameter. The details of the spectral clustering algorithm are in our supplementary materials. We average the corresponding feature $f_r, r = 1, 2, \ldots, t$ of each region $r$ belonging to each cluster $C_j$ as follows.

$$\hat{\mu}_j^I = \frac{1}{|C_j|} \sum_{r \in C_j} f_r^I, \qquad j = 1, 2, \ldots, K \tag{4}$$

where $f_r^I \in F(\mathbf{f}^I), r \in C_j$ are local features of image $I$ grouped into cluster $j$ through spectral clustering. $\hat{\boldsymbol{\mu}}^I = \{\hat{\mu}_j^I\}$ represents **semantic patterns** inside the image $I$. Fig. 4 visualizes some examples of $\hat{\mu}_j^I$. The whole process of semantic pattern extraction is shown in Alg. 1

### 4.3 Sample Selection with Semantic Patterns

Our main target of data selection is to make the distributions of selected samples diverse and representative in the level of local *semantic patterns* instead of the global feature level. This fine-grained strategy guarantees that our selected subset can cover rich local visual patterns represented by different pseudo-categories, which are crucial for detection and segmentation tasks.

To this end, we adopt a distance-based sampling strategy at the semantic pattern level. The detailed algorithm is shown in Alg. 2. Given an unlabeled image pool $\mathcal{I}$, this process starts from randomly selecting an initial image $I_0$ *i.e.* selecting all semantic patterns $\hat{\boldsymbol{\mu}}^{I_0}$ inside it. Then, we choose the next semantic pattern $\hat{\mu}_j^I$ inside image $I$ with probability in proportion to its squared distances from the nearest already selected semantic pattern (Eq. 5).

$$p(\hat{\mu}_j^I) \propto \min_{\hat{\mu} \in S_{\mathcal{K}}} \left[D(\hat{\mu}_j^I, \hat{\mu})\right]^2, I \in \mathcal{I}, j = 1, 2, \ldots, K \tag{5}$$

where $S_{\mathcal{K}}$ is the pool of all the already selected semantic patterns. When we choose a semantic pattern $\hat{\mu}_j^I$, all the semantic patterns $\hat{\boldsymbol{\mu}}^I$ inside the image $I$ that contains $\hat{\mu}_j^I$ are put into the selected pool $S_{\mathcal{K}}$. We use cosine distance for $D(\cdot, \cdot)$ as analyzed in the supplementary materials. This process continues until enough images have been selected. The selection only requires semantic patterns constructed from intermediate features offline beforehand. Consequently, only a *single-pass* model inference *without* any training or supervision is required in the entire data selection pipeline.

## 5 Experiments

We evaluate FreeSel on object detection (Sec. 5.2), semantic segmentation (Sec. 5.3), and image classification (Sec. 5.4). The results of FreeSel are *averaged over three independent selections with different random seeds*. Features are extracted by the same general pretrained model for all the tasks (Sec. 5.1). We make some analysis of our proposed pipeline and method in Sec. 5.5. Finally, we examine the roles of different modules inside FreeSel in Sec. 5.6. We refer readers to supplementary materials for more implementation details, results, and ablation studies.

### 5.1 General Model for Feature Extraction

We adopt DeiT-S [50] (path size 16×16) pretrained with the unsupervised framework DINO [5] on ImageNet [11] to extract features for data selection. The same pretrained model is used for all tasks.

**Algorithm 2: Distance-based Selection**

**Input:** all semantic patterns $\hat{\boldsymbol{\mu}}^I = \{\hat{\mu}_j^I\}$ for each image $I$, total annotation budget size $b$
**Output:** selected image pool $S_\mathcal{I}$

1   Initialize $S_\mathcal{I} = \{I_0\}$
    /* initialize with a random image $I_0$     */
2   Initialize $S_\mathcal{K} = \{\hat{\mu}_j^{I_0}, j = 1, \ldots, K\}$
    /* initialize selected semantic pattern pool with all semantic patterns in $I_0$  */
3   **repeat**
4      Sample $\hat{\mu}_j^I$ with probability $p(\hat{\mu}_j^I) \propto \min_{\hat{\mu} \in S_\mathcal{K}} \left[ D(\hat{\mu}_j^I, \hat{\mu}) \right]^2$
      /* sample next semantic pattern $\hat{\mu}_j^I$ with the distance-based probability.    */
5      $S_\mathcal{I} = S_\mathcal{I} \cup \{I\}$
      /* add image $I$ containing sampled $\hat{\mu}_j^I$ to selected image pool     */
6      $S_\mathcal{K} = S_\mathcal{K} \cup \{\hat{\mu}_j^I, j = 1, \ldots, K\}$
      /* add all semantic patterns in image $I$ to selected semantic pattern pool   */
7   **until** $|S_\mathcal{I}| = b$;

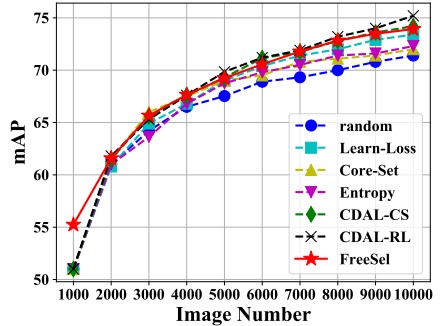

Figure 5: **Results on Object Detection:** The mAP on 100% training data is 77.43.

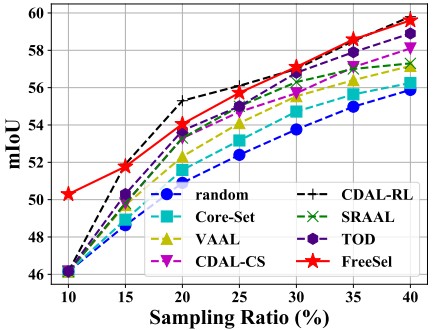

Figure 6: **Results on Semantic Segmentation:** The mIoU on 100% training data is 62.95.

FreeSel can also fit other frameworks as well, as shown in supplementary materials. We emphasize that this pretrained DeiT-S model is only applied to the data selection process. For the downstream tasks, we still train the convolutional task models from scratch in accordance with prior work.

## 5.2   Object Detection

**Dataset and Task Model.** We carry out experiments on PASCAL VOC [14]. In line with prior work [1, 57], we combine the training and validation sets of PASCAL VOC 2007 and 2012 as the training data pool with $16,551$ images. The performance of task model is evaluated on PASCAL VOC 2007 test set using *mAP* metric. We follow previous work [57, 1] to train an SSD-300 model [34] with VGG-16 backbone [46] on the selected samples. It reaches 77.43 mAP with $100\%$ training data.

**Results and Comparison.** We compare our performance with existing active learning methods (Fig. 5) for multiple sampling ratios. For fairness, we only include task-agnostic methods instead of those designed specifically for object detection [59, 8] which should naturally perform better. Results show that FreeSel outperforms most traditional pipeline methods and remains competitive with the best ones. Besides, all these previous methods require repetitive model training and batch selection on each target dataset separately, while FreeSel can efficiently select all samples in a single pass. Sec. 5.6 also shows that FreeSel can outperform other alternative general-purpose model baselines.

## 5.3   Semantic Segmentation

**Dataset and Task Model.** We use Cityscapes [9] dataset for semantic segmentation. This dataset is composed of 3,475 frames with pixel-level annotation of 19 object classes. We report the result using *mIoU* metric. We follow previous active learning research to apply DRN [58] model for this task. It reaches 62.95 mIoU with $100\%$ training data.

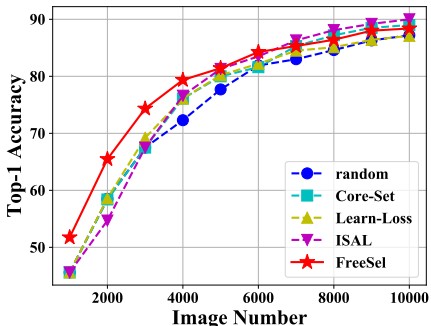
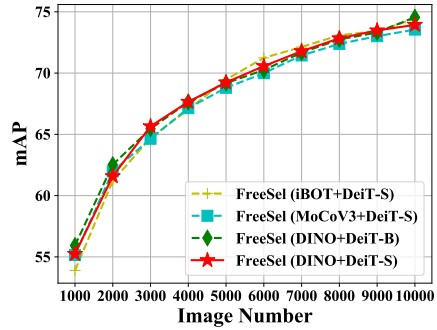

Figure 7: **Results on Image Classification:** The accuracy on 100% training data is 93.02%.

Figure 8: **Effect of Pretraining Methods:** Experiments are conducted on PASCAL VOC.

**Results.** We compare the performance of FreeSel with traditional active learning methods in Fig. 6. Given the domain gap between the pretraining dataset (*i.e.* ImageNet) and Cityscapes, it is quite challenging to utilize the general pretrained model for data selection. However, with much higher efficiency, FreeSel still beats most traditional pipeline methods in performance.

## 5.4 Image Classification

**Dataset and Task Model.** We use CIFAR-10 [29] datasets and ResNet-18 [21] model in line with prior work [35, 57]. CIFAR-10 contains 60,000 images with size 32×32 (50,000 for training and 10,000 for test) belonging to 10 categories. We report the results using *Top-1 Accuracy* metric. The model reaches 93.02% Top-1 Accuracy with 100% training data on CIFAR-10.

**Results.** We demonstrate the results of data selection methods in Fig. 7. Our performance is compared with traditional active learning methods as well. Since image classification focuses on global information, the advantage of semantic patterns cannot be fully demonstrated. However, with most sampling ratios, FreeSel still beats all its counterparts.

## 5.5 Analysis

**Time Efficiency Analysis.** Time efficiency of data selection is crucial for its practical use. Tab. 1 shows the comparison between FreeSel and other existing counterparts. The estimation is conducted on PASCAL VOC to choose 8,000 samples. We follow prior papers [1, 44, 57] to use SSD [34] as the task model (same as Sec. 5.2). The time is estimated on a single NVIDIA TITAN RTX GPU. Since FreeSel directly utilizes the publicly available pretrained model *instead of* training models separately for each dataset, only the feature extraction, semantic pattern construction, and data selection time should be considered, *i.e.* 285 seconds in total. In contrast, for other active learning methods, networks are trained repetitively on each dataset. We follow [57, 1] to set their initial set size and batch selection budget both as $1k$, so their model should be trained for 7 times over subsets of size $1k \sim 7k$ to select 8,000 samples. These previous methods have similar time efficiency, requiring about 42 hours in total. They also need to wait for the oracle for ground-truth labels after selecting each batch of data. Based on the above information, our method can be **530x faster** than prior work.

**Single-Pass Data Selection.** Unlike prior active learning methods, FreeSel follows the new pipeline to select all the data samples in a single pass. This allows for great practical use. Firstly, it makes our method free of a random initial set. For one thing, FreeSel can bring performance gain in the lowest sampling ratio. This is beneficial in practice when the annotation budget is extremely low. For another thing, FreeSel would not suffer from the imbalanced initial set. As discussed in [47], low-quality initial sets would hurt the performance of prior active learning work significantly. Secondly, FreeSel simplifies the active learning pipeline from the iterative *model training→batch data selection→batch annotation→ ⋯⋯* to a single-pass *data selection→data annotation*, which saves notable efforts in the management, communication, and coordination of traditional sequential steps.

**Introduction of Pretrained Model.** Our proposed pipeline introduces a pretrained model (Fig. 1b) to satisfy the three principles of our new pipeline. Since the pretraining is not designed specifically for the data selection, directly using a publicly available model would not lead to extra time-cost or expense. According to Sec. 3, it is non-trivial to improve the efficiency of active learning with a

pretrained model. We further show that our great performance does not come from the pretrained model in Sec. 5.6.

**Effect of Different Pretraining Algorithms.** In this part, we pay attention to the effect of pretraining on the final performance of FreeSel. In addition to DeiT-S [50] pretrained with DINO framework [5] in Sec. 5.1, we also adopt two alternative pretraining frameworks MoCoV3 [7] and iBOT [63] as well as a larger DeiT-B model [50]. Those different pretrained models are applied to the data selection on PASCAL VOC dataset [14]. Same as Sec. 5.2, we train an SSD-300 model [34] on the selected samples for the object detection task. Fig. 8 demonstrates that FreeSel with different pretrained models for data selection only has marginal differences in the performance of the downstream object detection task. This result verifies that FreeSel can widely fit different pretraining algorithms. The great performance of data selection comes from our carefully designed modules in FreeSel instead of the strong representative ability of some specific pretrained models.

## 5.6 Ablation Study

We delve into different parts of our method. Firstly, we analyze the contribution of each module inside FreeSel to the final performance. Then, the role of the pretrained DeiT-S model is also discussed.

**Each Module Contribution.** Starting from the failure case in Fig. 2, modules of FreeSel are added to it one by one. Tab. 2 demonstrates the step-by-step performance improvement. This experiment is conducted on PASCAL VOC in the same setting as Sec. 5.2. The following three modules are analyzed. More quantitative analysis of hyper-parameters is available in the supplementary materials.

- **Feature Extraction Manner:** In Sec. 3, the global feature of [CLS] token is directly used. We replace it with the proposed semantic patterns defined in Eq. 4.

- **Attention Filter:** We apply the attention filter in Eq. 2 to filter local features.

- **Selection Strategy:** Apart from the distance-based sampling in Eq. 5, we consider the alternative farthest-distance-sampling (FDS) *w.r.t.* semantic patterns, which is theoretically justified in [44] as an approximation of K-Centers. It chooses the next semantic pattern farthest from the nearest selected one as $\hat{\mu}_j^I = arg\max_{\hat{\mu}_j^I} \min_{\hat{\mu} \in s_\mathcal{K}} d(\hat{\mu}_j^I, \hat{\mu})$.

The failure case of Core-Set on off-the-shelf features is shown in the first line of Tab. 2. Then, we extract features by constructing semantic patterns ($K = 5$) without applying the attention filter in the second line. It improves notably compared with the first line because the semantic patterns can represent useful local information important for object detection. However, it is only slightly better than random selection since the semantic patterns are dominated by local noisy information in this stage. We apply attention ratio $\tau = 0.5$ (Eq. 2) in the third line of the table, and the performance gets improved again. Finally, the FDS selection strategy is replaced by the distance-based probability sampling in Eq. 5. It provides extra performance gain because it would select more representative samples with fewer corner cases.

Table 2: **Module Contribution:** We discuss the contribution of each module inside FreeSel. SP means semantic pattern. Experiments are conducted on PASCAL VOC.

| Feature | Filter | Select | Image Number | | |
|---------|--------|--------|------|------|------|
| | | | 3k | 5k | 7k |
| global | ✗ | FDS | 60.59 | 66.65 | 70.30 |
| SP | ✗ | FDS | 64.15 | 68.22 | 70.42 |
| SP | $\tau = 0.5$ | FDS | 64.45 | 68.49 | 71.35 |
| SP | $\tau = 0.5$ | Prob. | **65.66** | **69.24** | **71.79** |
| *random sampling* | | | 64.21 | 67.53 | 69.32 |

**Role of Pretrained Model.** There is a concern that the performance gain of FreeSel comes from the great representation ability of the pretrained vision transformer for data selection instead of our designed method. About this, we conduct an ablation study on CIFAR-10 in the same setting as Sec. 5.4. We equip Core-Set [44] and Learn-Loss [57] with the same pretrained network for data selection, *i.e.* DeiT-S [50] pretrained with DINO [5]. During the data selection period, pretrained DeiT-S is finetuned supervisedly to select data samples with Core-Set and Learn-Loss algorithms. After selection, we still train ResNet-18 [21] over the selected samples from scratch. In Fig. 9, this pretrained DeiT-S damages the performance of Core-Set and Learn-Loss. A potential explanation comes from the differences in the representation spaces of pretrained DeiT and from-scratch ResNet. The samples selected by DeiT-S with Core-Set and Learn-Loss algorithms may not be suitable for the

from-scratch training of the ResNet-18 model. This reflects that our performance gain does not come from the use of pretrained DeiT-S. Instead, the proposed FreeSel method plays an important role.

**General-Purpose Model Baselines.** To further disentangle the roles of the general-purpose model and our designed FreeSel framework, we compare FreeSel with the following baselines which can also select a subset from the data pool using the general-purpose models. **1) K-Means:** We perform the K-Means algorithm on the global features extracted by the pretrained DeiT-S

Table 3: **Baselines Using General-Purpose Model:** We compare FreeSel with other baselines using the general-purpose model. Experiments are conducted on PASCAL VOC object detection task.

| Methods | Pretrained Model | Image Number | | |
|---|---|---|---|---|
| | | 3k | 5k | 7k |
| K-Means | DeiT-S (DINO) | 64.85 | 68.05 | 71.50 |
| Inconsistency | DeiT-S (DINO) | 63.29 | 67.65 | 71.35 |
| Entropy | DeiT-S (supervised) | 56.33 | 66.03 | 69.72 |
| FreeSel | DeiT-S (DINO) | **65.66** | **69.24** | **71.79** |

model [50, 5], choosing the sample closest to each cluster center. **2) Inconsistency:** We select the most difficult samples based on the inconsistency of multiple-time model predictions. To measure the inconsistency, we perform data augmentations (RandAugment [10]) to generate 10 different augmented copies for each image and calculate the average pairwise distances of global features between these copies extracted by the pretrained DeiT-S model [50, 5]. We select data samples by the order of inconsistency. **3) Entropy:** We select the most ambiguous samples based on the classification uncertainty of the pretrained model. Since the classification score is required, we adopt the DeiT-S model [50] pretrained on ImageNet in a supervised manner and measure the uncertainty with the entropy of classification scores. We select data samples by the order of entropy. Experiments are conducted on object detection task in the same settings as Sec. 5.2. Tab. 3 shows that all the above baselines perform notably worse than FreeSel, especially with low sampling ratios. This reflects the importance of our proposed FreeSel algorithm. Trivial utilization of a general-purpose model would not lead to great performance of data selection.

## 6   Conclusion and Limitations

The main goal of this paper is to enable a free data selection pipeline by proposing a novel pipeline with three key principles: generality, efficiency, and non-supervision. We verify its feasibility by designing the first method FreeSel following this new pipeline. Through a single-pass model inference, the semantic patterns are constructed based on the intermediate features of a general pretrained model, over which a distance-based selection strategy is performed to find the most diverse and informative data samples. Our method outperforms most existing counterparts with remarkably superior efficiency on different tasks including detection, segmentation, and classification.

We realize that FreeSel cannot beat all the other data selection methods in current stage due to the absence of training on the target datasets. Never-

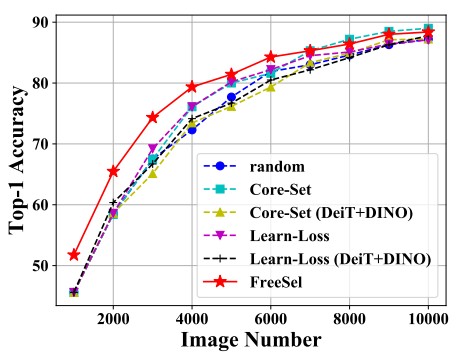

Figure 9: **Effect of Pretraining Methods:** Experiments are conducted on PASCAL VOC.

theless, this direction matters in boosting the training of downstream models without any extra time and cost on the shoulders of existing general pretrained models. It gains more significance given the current landscape dominated by large foundation models pretrained on multi-modality data [40, 15], which we believe can help to extend our method to a wide range of domains and modalities.

**Acknowledgement.** This work is partially supported by Berkeley DeepDrive.

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

In this supplementary material, we first explain the details of spectral clustering algorithm (Sec. 4.2) in Sec. A. We then analyze the sensitivity of FreeSel to the values of hyperparameters in Sec. B. Finally, implementation details of our experiments are explained in Sec. C.

## A Spectral Clustering Algorithm

In this section, we explain the spectral clustering algorithm [38, 55] in the semantic pattern extraction process for each image $I$ (Sec. 4.2 and Alg. 1). The detailed spectral clustering algorithm is shown in Alg. 3. This spectral clustering algorithm should be inserted into line 7 of Alg. 1.

To justify the use of spectral clustering algorithm for semantic pattern extraction, we also try another alternative which directly performs K-Means *w.r.t.* the local features $f_r^I, r = 1, 2, \ldots, t$ to divide the $t$ regions of image $I$ into $K$ clusters without using the patch token attention to achieve the same feature clustering goal in Sec. 4.2. Tab. 4 shows the comparison between spectral clustering and K-Means. Interestingly, these two feature clustering strategies lead to similar data selection performance on PASCAL VOC [14] object detection task. However, spectral clustering is stably superior when selecting data samples for Cityscapes [9] semantic segmentation task. We attribute this difference to the large domain gap between Cityscapes dataset and ImageNet dataset [11]. The DeiT-S model pretrained on ImageNet may extract local features with weaker discriminative ability from images inside Cityscapes dataset. Since spectral clustering algorithm depends less on the feature quality, it can bring better performance than direct K-Means over intermediate local features on Cityscapes.

---

**Algorithm 3: Spectral Clustering**

**Input:** Similarity matrix between patches $\widehat{\mathbf{pa}}^I = \left[\widehat{pa}_{ij}\right]_{i,j=1,2,\ldots,t}$, semantic pattern number $K$

**Output:** Clusters $C_j^I, j = 1, 2, \ldots, K$, where each region $r = 1, 2, \ldots, t$ of image $I$ belongs to a unique $C_j^I$.

1 Derive the symmetric adjacent matrix $\mathbf{A}$ from $\widehat{\mathbf{pa}}^I$:

$$\mathbf{A} = (\widehat{\mathbf{pa}}^I + \widehat{\mathbf{pa}}^{I^T})/2, \qquad \mathbf{A} \in \mathbb{R}^{t \times t}$$

2 Derive the diagonal degree matrix $\mathbf{D}$:

$$\mathbf{D}_{ij} = \begin{cases} \sum_{l=1}^t \mathbf{A}_{il} & i = j \\ 0 & i \neq j \end{cases}, \qquad \mathbf{D} \in \mathbb{R}^{t \times t}$$

3 Calculate the normalized Laplacian matrix $\mathbf{L}$:

$$\mathbf{L} = \mathbf{D}^{-\frac{1}{2}}(\mathbf{D} - \mathbf{A})\mathbf{D}^{-\frac{1}{2}}, \qquad \mathbf{L} \in \mathbb{R}^{t \times t}$$

4 Obtain the $K$ eigenvectors $v_l, l = 1, 2, \ldots, K$ corresponding to the $K$ smallest eigenvalues $\sigma_l, l = 1, 2, \ldots, K$ of matrix $\mathbf{L}$.

5 Compose the matrix $\mathbf{V}$ based on the $K$ eigenvectors

$$\mathbf{V} = [v_1, v_2, \ldots, v_K], \mathbf{V} \in \mathbf{R}^{t \times K}$$

6 Denote $u_i^T$ as the $i$-th row of $\mathbf{V}$, $i = 1, 2, \ldots, t$

7 Normalize each row of $\mathbf{V}$: $\hat{u}_i = u_i / \sqrt{\sum_{j=1}^K u_{i,j}^2}$

8 Perform K-Means to divide $\hat{u}_i, i = 1, 2, \ldots, t$ into $K$ clusters $C_j^I, j = 1, 2, \ldots, K$:

$$\{C_j^I\}_{j=1,2,\ldots,K} = KMeans(\{\hat{u}_i\}_{i=1,2,\ldots,t})$$

---

## B Sensitivity to Hyperparameters

In this part, we analyze the sensitivity of our FreeSel to some hyperparameters including the maintenance ratio $\tau$ in the attention filter (Eq. 2), the semantic pattern number $K$ (Eq. 4), the neighborhood threshold $d_0$ (Eq. 3), the distance function $D(\cdot, \cdot)$ (Eq. 5), and pretraining manner for the general

Table 4: **Effect of Feature Clustering Strategies:** We compare spectral clustering and K-Means for feature clustering. Experiments are conducted on PASCAL VOC object detection task and Cityscapes semantic segmentation task.

(a) **Performance on PASCAL VOC Object Detection Task:** The task model is SSD-300 [34].

| Feature Clustering | Image Number | | |
|---|---|---|---|
| | $3k$ | $5k$ | $7k$ |
| K-Means | 65.35 | **69.43** | 71.76 |
| Spectral Clustering | **65.66** | 69.24 | **71.79** |

(b) **Performance on Cityscapes Semantic Segmentation Task:** The task model is DRN [58].

| Feature Clustering | Sampling Ratio | | |
|---|---|---|---|
| | 15% | 25% | 35% |
| K-Means | 51.43 | 54.84 | 57.96 |
| Spectral Clustering | **51.77** | **55.72** | **58.58** |

Table 5: **Sensitivity to Hyperparameters:** $\tau, K, d_0, D(\cdot, \cdot)$ separately denote the maintenance ratio, semantic pattern number, neighborhood threshold, and distance function. Experiments are conducted on the PASCAL VOC object detection task.

| $\tau$ | **K** | $d_0$ | $D(\cdot, \cdot)$ | **Pretraining** | Image Number | | |
|---|---|---|---|---|---|---|---|
| | | | | | $3k$ | $5k$ | $7k$ |
| 0.3 | | | | | 65.22 | 69.00 | 70.69 |
| 0.5 | 5 | 2 | cos. | *unsupervised* | **65.66** | 69.24 | **71.79** |
| 0.7 | | | | | 64.77 | 69.33 | 71.64 |
| 0.5 | 1 | 2 | cos. | *unsupervised* | 64.90 | 69.01 | 71.02 |
| | 10 | | | | 65.21 | 69.13 | 71.50 |
| 0.5 | 5 | 1 | cos. | *unsupervised* | 65.48 | 68.73 | 71.39 |
| | | 3 | | | 65.37 | 69.41 | 71.74 |
| 0.5 | 5 | 2 | euc. | *unsupervised* | 64.77 | **69.42** | 71.31 |
| 0.5 | 5 | 2 | cos. | *supervised* | 64.40 | 68.82 | 71.43 |

model. Experiments are conducted on the object detection task, where samples are selected from PASCAL VOC dataset and SSD-300 is the downstream task model in the same settings as Sec. 5.2. Results are shown in Tab. 5.

**Maintenance Ratio $\tau$ (Eq. 2)** Maintenance ratio $\tau$ notably affects the final performance of FreeSel. Too low ratios lead to the ignorance of some crucial local visual patterns, while too high ratios introduce some harmful noisy information to the semantic patterns. Thus, a moderate attention ratio plays an important role in the high performance of FreeSel.

**Semantic Pattern Number $K$ (Eq. 4)** When $K = 1$, the performance is hurt since semantic patterns degrade to global features in this case. When $K = 10$, a slight performance drop may be witnessed in comparison with $K = 5$.

**Neighborhood Threshold $d_0$ (Eq. 3)** When $d_0 = 1$, the neighborhood is too small to represent the relationship between nearby regions. When $d_0 = 3$, the performance is a little worse than $d_0 = 2$. We think each region feature mainly interacts with nearby regions with distance $d \leq 2$.

**Distance Function $D(\cdot, \cdot)$ (Eq. 5)** We find the cosine distance can lead to better performance than Euclidean distance. This result shows that the directions of local feature vectors are important to reflect the diversity of local visual patterns.

**Pretraining Manner** Instead of using the unsupervised pretraining framework DINO [5], we also try the DeiT-S model [50] pretrained in a supervised manner on ImageNet [30]. Results show a performance drop with supervised pretraining. We think this is because supervised pretraining introduces some biases of categories to the pretrained model.

## C  Implementation Details

### C.1  Object Detection Implementation

#### C.1.1  Implementation of FreeSel

We set attention ratio $\tau = 0.5$ and semantic pattern number $K = 5$. The input images are resized to $224 \times 224$ when fed into the pretrained DeiT-S [50] model in the data selection process.

#### C.1.2  Implementation of Task Model

The implementation of task model is same as previous active learning research [57, 1]. The SSD-300 model [34] with VGG-16 [46] backbone is adopted for this experiment. The model is implemented based on mmdetection [3]. We follow [57, 1] to train the model for 300 epochs with batch size 32 using SGD optimizer (momentum 0.9). The initial learning rate is 0.001, which decays to 0.0001 after 240 epochs.

### C.2  Semantic Segmentation Implementation

#### C.2.1  Implementation of FreeSel

The input images are resized to $448 \times 224$ in line with their original aspect ratios when fed into the pretrained DeiT-S [50] model in the data selection process. Same as object detection, we set attention ratio $\tau = 0.5$ and semantic pattern number is doubled to $K = 10$ in line with the doubled input size compared to object detection task.

#### C.2.2  Implementation of Task Model

We follow prior active learning work [47, 24] to apply DRN [58] model [4] for semantic segmentation task. The model is trained for 50 epochs with batch size 8 and learning rate 5e-4 using Adam optimizer [28].

### C.3  Image Classification Implementation

#### C.3.1  Implementation of FreeSel

We follow previous tasks to set attention ratio $\tau = 0.5$. Since image classification depends less on local information, we directly set the semantic pattern number $K = 1$. The input images are resized to $224 \times 224$ when fed into the pretrained DeiT-S [50] model in the data selection process.

#### C.3.2  Implementation of Task Model

We follow [57, 35] to use ResNet-18 [21] classification model in this task, which is implemented based on mmclassification [5]. The model is trained for 200 epochs with batch size 128 using an SGD optimizer (momentum 0.9, weight decay 5e-4). The initial learning rate is 0.1, which decays to 0.01 after 160 epochs. We apply standard data augmentation to the training including 32×32 size random crop from 36×36 zero-padded images and random horizontal flip.

---

[3]https://github.com/open-mmlab/mmdetection
[4]https://github.com/fyu/drn
[5]https://github.com/open-mmlab/mmclassification

