# Supplementary Materials for the Paper "Towards Free Data Selection with General-Purpose Models"

In this supplementary material, we first explain the details of spectral clustering algorithm in Sec. 4.2 of our main paper in Sec. A. We then discuss the effect of different pretraining frameworks and models in Sec. B. We also analyze the sensitivity of FreeSel to the values of hyperparameters in Sec. C. Besides, FreeSel is compared with other intuitive baselines using the general-purpose model in Sec. D. Finally, implementation details of our experiments are explained in Sec. E. Our code will be made publicly available.

## A    Spectral Clustering Algorithm

In this section, we explain the spectral clustering algorithm [14, 18] in the semantic pattern extraction process for each image $I$ (Sec. 4.2 and Alg. 1 of our main paper). The detailed spectral clustering algorithm is shown in Alg. 1. This spectral clustering algorithm should be inserted into line 7 of Alg. 1 in our main paper.

To justify the use of spectral clustering algorithm for semantic pattern extraction, we also try another alternative which directly performs K-Means *w.r.t.* the local features $f_r^I, r = 1, 2, \ldots, t$ to divide the $t$ regions of image $I$ into $K$ clusters without using the patch token attention to achieve the same feature clustering goal in Sec. 4.2 of our main paper. Tab. 1 shows the comparison between spectral clustering and K-Means. Interestingly, these two feature clustering strategies lead to similar data selection performance on PASCAL VOC [7] object detection task. However, spectral clustering is stably superior when selecting data samples for Cityscapes [4] semantic segmentation task. We attribute this difference to the large domain gap between Cityscapes dataset and ImageNet dataset [6]. The DeiT-S model pretrained on ImageNet may extract local features with weaker discriminative ability from images inside Cityscapes dataset. Since spectral clustering algorithm depends less on the feature quality, it can bring better performance than direct K-Means over intermediate local features on Cityscapes.

## B    Effect of Pretraining Methods

In this part, we pay attention to the effect of pretraining on the final performance of FreeSel. In addition to the DeiT-S model [17] pretrained with DINO framework [2] in our main paper, we also adopt two alternative pretraining frameworks MoCoV3 [3] and iBOT [21] as well as a larger DeiT-B model [17]. Those different pretrained models are applied to the data selection on PASCAL VOC dataset [7]. Same as Sec. 5.2 of our main paper, we train an SSD-300 model [12] on the selected samples for the object detection task. Fig. 1 demonstrates that FreeSel with different pretrained models for data selection only has marginal differences in the performance of the downstream object detection task. This result verifies that FreeSel can widely fit different pretraining algorithms. The great performance of data selection comes from our carefully designed modules in FreeSel instead of the strong representative ability of some specific pretrained models.

Submitted to 37th Conference on Neural Information Processing Systems (NeurIPS 2023). Do not distribute.

**Algorithm 1: Spectral Clustering**

---

**Input:** Similarity matrix between patches $\widehat{\mathbf{pa}}^I = \left[\widehat{pa}_{ij}\right]_{i,j=1,2,\dots,t}$, semantic pattern number $K$

**Output:** Clusters $C_j^I, j = 1, 2, \dots, K$, where each region $r = 1, 2, \dots, t$ of image $I$ belongs to a unique $C_j^I$.

**1** Derive the symmetric adjacent matrix $\mathbf{A}$ from $\widehat{\mathbf{pa}}^I$:

$$\mathbf{A} = (\widehat{\mathbf{pa}}^I + \widehat{\mathbf{pa}}^{IT})/2, \qquad \mathbf{A} \in \mathbb{R}^{t \times t}$$

**2** Derive the diagonal degree matrix $\mathbf{D}$:

$$\mathbf{D}_{ij} = \begin{cases} \sum_{l=1}^t \mathbf{A}_{il} & i = j \\ 0 & i \neq j \end{cases}, \qquad \mathbf{D} \in \mathbb{R}^{t \times t}$$

**3** Calculate the normalized Laplacian matrix $\mathbf{L}$:

$$\mathbf{L} = \mathbf{D}^{-\frac{1}{2}}(\mathbf{D} - \mathbf{A})\mathbf{D}^{-\frac{1}{2}}, \qquad \mathbf{L} \in \mathbb{R}^{t \times t}$$

**4** Obtain the $K$ eigenvectors $v_l, l = 1, 2, \dots, K$ corresponding to the $K$ smallest eigenvalues $\sigma_l, l = 1, 2, \dots, K$ of matrix $\mathbf{L}$.

**5** Compose the matrix $\mathbf{V}$ based on the $K$ eigenvectors

$$\mathbf{V} = [v_1, v_2, \dots, v_K], \mathbf{V} \in \mathbf{R}^{t \times K}$$

**6** Denote $u_i^T$ as the $i$-th row of $\mathbf{V}$, $i = 1, 2, \dots, t$

**7** Normalize each row of $\mathbf{V}$: $\hat{u}_i = u_i / \sqrt{\sum_{j=1}^K u_{i,j}^2}$

**8** Perform K-Means to divide $\hat{u}_i, i = 1, 2, \dots, t$ into $K$ clusters $C_j^I, j = 1, 2, \dots, K$:

$$\{C_j^I\}_{j=1,2,\dots,K} = KMeans(\{\hat{u}_i\}_{i=1,2,\dots,t})$$

---

Table 1: **Effect of Feature Clustering Strategies:** We compare spectral clustering and K-Means for feature clustering. Experiments are conducted on PASCAL VOC object detection task and Cityscapes semantic segmentation task.

(a) **Performance on PASCAL VOC Object Detection Task:** The task model is SSD-300 [12].

| Feature Clustering | Image Number | | |
| --- | --- | --- | --- |
| | 3k | 5k | 7k |
| K-Means | 65.35 | **69.43** | 71.76 |
| Spectral Clustering | **65.66** | 69.24 | **71.79** |

(b) **Performance on Cityscapes Semantic Segmentation Task:** The task model is DRN [20].

| Feature Clustering | Sampling Ratio | | |
| --- | --- | --- | --- |
| | 15% | 25% | 35% |
| K-Means | 51.43 | 54.84 | 57.96 |
| Spectral Clustering | **51.77** | **55.72** | **58.58** |

## C  Sensitivity to Hyperparameters

In this part, we analyze the sensitivity of our FreeSel to some hyperparameters including the maintenance ratio $\tau$ in the attention filter (Eq. 2 of our main paper), the semantic pattern number $K$ (Eq. 4 of our main paper), the neighborhood threshold $d_0$ (Eq. 3 of our main paper), the distance function $D(\cdot, \cdot)$ (Eq. 5 of our main paper), and pretraining manner for the general model. Experiments are conducted on object detection task, where samples are selected from PASCAL VOC dataset and SSD-300 is the downstream task model in the same settings as Sec. 5.2 of our main paper. Results are shown in Tab. 2.

**Maintenance Ratio $\tau$ (Eq. 2 of main paper)**  Maintenance ratio $\tau$ notably affects the final performance of FreeSel. Too low ratios lead to the ignorance of some crucial local visual patterns, while too high ratios introduce some harmful noisy information to the semantic patterns. Thus, a moderate attention ratio plays an important role in the high performance of FreeSel.

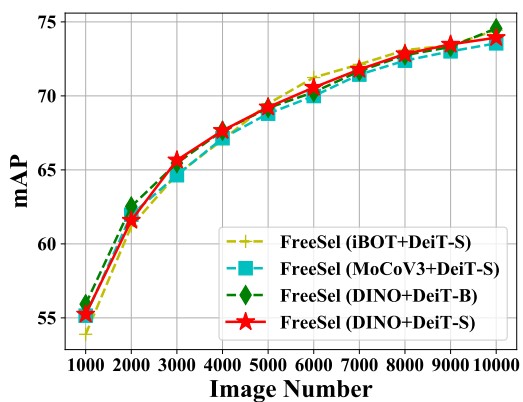

Figure 1: **Effect of Pretraining Methods:** We evaluate the performance of FreeSel with different pretrained models. The red stars denote the results in our main paper. Experiments are conducted on PASCAL VOC dataset with SSD-300 as the object detection task model.

Table 2: **Sensitivity to Hyperparameters:** $\tau, K, d_0, D(\cdot, \cdot)$ separately denote the maintenance ratio, semantic pattern number, neighborhood threshold, and distance function. Experiments are conducted on PASCAL VOC object detection task.

| $\tau$ | $K$ | $d_0$ | $D(\cdot, \cdot)$ | Pretraining | Image Number | | |
|--------|-----|-------|-------------------|-------------|------|------|------|
| | | | | | $3k$ | $5k$ | $7k$ |
| 0.3 | | | | | 65.22 | 69.00 | 70.69 |
| 0.5 | 5 | 2 | cos. | *unsupervised* | **65.66** | 69.24 | **71.79** |
| 0.7 | | | | | 64.77 | 69.33 | 71.64 |
| 0.5 | 1 | 2 | cos. | *unsupervised* | 64.90 | 69.01 | 71.02 |
| | 10 | | | | 65.21 | 69.13 | 71.50 |
| 0.5 | 5 | 1 | cos. | *unsupervised* | 65.48 | 68.73 | 71.39 |
| | | 3 | | | 65.37 | 69.41 | 71.74 |
| 0.5 | 5 | 2 | euc. | *unsupervised* | 64.77 | **69.42** | 71.31 |
| 0.5 | 5 | 2 | cos. | *supervised* | 64.40 | 68.82 | 71.43 |

**Semantic Pattern Number** $K$ **(Eq. 4 of main paper)** When $K = 1$, the performance is hurt since semantic patterns degrade to global features in this case. When $K = 10$, a slight performance drop may be witnessed in comparison with $K = 5$.

**Neighborhood Threshold** $d_0$ **(Eq. 3 of main paper)** When $d_0 = 1$, the neighborhood is too small to represent the relationship between nearby regions. When $d_0 = 3$, the performance is a little worse than $d_0 = 2$. We think each region feature mainly interacts with nearby regions with distance $d \leq 2$.

**Distance Function** $D(\cdot, \cdot)$ **(Eq. 5 of main paper)** We find the cosine distance can lead to better performance than Euclidean distance. This result shows that the directions of local feature vectors are important to reflect the diversity of local visual patterns.

**Pretraining Manner** Instead of using the unsupervised pretraining framework DINO [2], we also try the DeiT-S model [17] pretrained in a supervised manner on ImageNet [11]. Results show a performance drop with supervised pretraining. We think this is because supervised pretraining introduces some biases of categories to the pretrained model.

Table 3: **Baselines Using General-Purpose Model:** We compare FreeSel with other baselines using the general-purpose model. Experiments are conducted on PASCAL VOC object detection task.

| Methods | Pretrained Model | Image Number | | |
|---|---|---|---|---|
| | | $3k$ | $5k$ | $7k$ |
| K-Means | DeiT-S (DINO) | 64.85 | 68.05 | 71.50 |
| Inconsistency | DeiT-S (DINO) | 63.29 | 67.65 | 71.35 |
| Entropy | DeiT-S (supervised) | 56.33 | 66.03 | 69.72 |
| FreeSel | DeiT-S (DINO) | **65.66** | **69.24** | **71.79** |

## D   Baselines Using General-Purpose Model

To further disentangle the roles of the general-purpose model and our designed FreeSel framework, we compare FreeSel with the following baselines which can also select a subset from the data pool using the general-purpose models.

- **K-Means:** We perform the K-Means algorithm on the global features extracted by the DeiT-S model [17] pretrained with DINO [2]. The cluster number equals to the annotation budget size, and we choose the sample closest to each cluster center.

- **Inconsistency:** We select the most difficult samples based on the inconsistency of multiple-time model predictions. To measure the inconsistency, we perform data augmentations (RandAugment [5]) to generate 10 different augmented copies for each image. The inconsistency is measured by calculating the average pairwise distances of global features between these copies extracted by the DeiT-S model [17] pretrained with DINO [2]. We select data samples by the order of inconsistency.

- **Entropy:** We select the most ambiguous samples based on the classification uncertainty of the pretrained model. Since the classification score is required, we adopt the DeiT-S model [17] pretrained on ImageNet in a supervised manner and measure the uncertainty with the entropy of classification scores. We select data samples by the order of entropy.

Experiments are conducted on object detection task, where samples are selected from PASCAL VOC dataset and SSD-300 is the downstream task model in the same settings as Sec. 5.2 of our main paper. Tab. 3 shows that all the above baselines perform notably worse than FreeSel, especially with low sampling ratios. This reflects the importance of our proposed FreeSel algorithm. Trivial utilization of a general-purpose model would not lead to great performance of data selection.

## E   Implementation Details

### E.1   Object Detection Implementation

#### E.1.1   Implementation of FreeSel

We set attention ratio $\tau = 0.5$ and semantic pattern number $K = 5$. The input images are resized to $224 \times 224$ when fed into the pretrained DeiT-S [17] model in the data selection process.

#### E.1.2   Implementation of Task Model

The implementation of task model is same as previous active learning research [19, 1]. The SSD-300 model [12] with VGG-16 [15] backbone is adopted for this experiment. The model is implemented based on mmdetection [1]. We follow [19, 1] to train the model for 300 epochs with batch size 32 using SGD optimizer (momentum 0.9). The initial learning rate is 0.001, which decays to 0.0001 after 240 epochs.

---

[1]https://github.com/open-mmlab/mmdetection

### E.2    Semantic Segmentation Implementation

### E.2.1    Implementation of FreeSel

The input images are resized to $448 \times 224$ in line with their original aspect ratios when fed into the pretrained DeiT-S [17] model in the data selection process. Same as object detection, we set attention ratio $\tau = 0.5$ and semantic pattern number is doubled to $K = 10$ in line with the doubled input size compared to object detection task.

### E.2.2    Implementation of Task Model

We follow prior active learning work [16, 9] to apply DRN [20] model [2] for semantic segmentation task. The model is trained for 50 epochs with batch size 8 and learning rate 5e-4 using Adam optimizer [10].

### E.3    Image Classification Implementation

### E.3.1    Implementation of FreeSel

We follow previous tasks to set attention ratio $\tau = 0.5$. Since image classification depends less on local information, we directly set the semantic pattern number $K = 1$. The input images are resized to $224 \times 224$ when fed into the pretrained DeiT-S [17] model in the data selection process.

### E.3.2    Implementation of Task Model

We follow [19, 13] to use ResNet-18 [8] classification model in this task, which is implemented based on mmclassification [3]. The model is trained for 200 epochs with batch size 128 using SGD optimizer (momentum 0.9, weight decay 5e-4). The initial learning rate is 0.1, which decays to 0.01 after 160 epochs. We apply standard data augmentation to the training including 32×32 size random crop from 36×36 zero-padded images and random horizontal flip.

---

[2]https://github.com/fyu/drn
[3]https://github.com/open-mmlab/mmclassification