# OpenReview forum: "Towards Free Data Selection with General-Purpose Models"
_NeurIPS.cc/2023/Conference — NeurIPS 2023 poster_

### Official Review · Reviewer_yLeE · 2023-07-05

**Soundness:** 3 good
**Presentation:** 3 good
**Contribution:** 2 fair
**Rating:** 3
**Confidence:** 4

**Summary:**

This paper presents a data selection methods without training, so saving lots of time than previous methods. By a publicly available pretrained model, FreeSel can select data from datasets with a single-pass inference, without need for additional training. This method uses comparison between semantic patterns of samples with distance-based sampling. Experiments on various tasks including classification, segmentation and detection verify the motivation.

**Strengths:**

1. The proposed method requires no training to select useful examples, saving lots of time.
2. The experimental performance shows the priority about this method.

**Weaknesses:**

1. The method is plain, to compare the similarities among input sample features. Though saving time, it cannot outperform other methods.
2. The downstream experiments are still using detection and segmentation models and backbones 5 or 6 years ago, the reviewer would like to see results on COCO, ADE20k and backbone like ViT, or DETR detectors.
3. The active learning which select samples from already annotated datasets is meaningless, more experiments about using FreeSel to select data from web or unannotated images are much more significant to show the effectiveness of this method.

**Questions:**

Please see weaknesses. If authors address them, reviewer would like to change the rating.

**Limitations:**

Yes.

---

> ### Author Rebuttal · Authors · 2023-08-09
>
> Thank you for your efforts to improve the quality of our papers. We thank you for pointing out your concerns, which we believe have been carefully clarified and addressed, as follows.
>
>
> **Q1: [Our method is plain and cannot outperform all active learning algorithms.]**
>
> Thank you for the question. We answer from two aspects: novelty and performance.
>
> > Regarding our novelty:
> - The main novelty of our work lies in the brand-new data selection pipeline. Existing active learning methods need repetitive model training and batch data selection, while other subset selection algorithm also requires to train the model on a small initial subset of the target dataset. The problem is that training models is costly, and the size of subset for training directly affects their performance of data selection. In contrast, our paper designs a novel pipeline to directly exploit general pretrained models to select all the samples from a large data pool in one go. In this case, no training on the target dataset is needed, and data selection time goes down from days to minutes, almost negligible, which means large-scale data selection becomes doable.
> - Although our method is notably straightforward without intricate modules, we believe it's unnecessary to inject excessively complex designs to artificially make it "looks novel". In contrast, it is always desirable that a simple method can finish the task well, especially given that our main novelty lies in the pipeline design. This is verified by our experiment results. If a simple method can rival well-designed existing algorithms in performance, it unquestionably becomes a compelling choice.
>
> > Regarding our performance:
> - In comparison with SOTA active learning algorithms, we can outperform most of them but underperform some of them in the late stage. Existing active learning methods train their models on the selected samples after each batch selection step. As a result, as the sample number increases, their models have been trained on a large subset of the target dataset, so it should naturally fit the target domain well. In contrast, our method makes use of general pretrained knowledge without any domain-specific training regarding the target dataset. That's why existing methods can fit the domain of the targeted dataset better after seeing many training samples in a later stage. This is reasonable, and our model therefore has wider applicability, such as scenarios where data is scarce or resources are severely constrained. We believe that the early stage is also critical as training budgets and data vary.
> - Nonetheless, we assert that performance isn't the sole metric of consideration. We aim to develop a "free" data selection algorithm. We can directly exploit public pretrained model and only need negligible extra efforts for data selection, reducing the data selection time from days to minutes as well as simplifying the data selection pipeline greatly (Line 249-257). In many scenarios, such as selecting data from an extremely large data pool, we believe this advantage of "free" far outweighs minor disparities in performance.
>
> **Q2: [Results on more datasets and backbones.]**
>
> The networks and datasets for our experiments follow the popular settings of existing active learning papers so that we can make fair comparisons with them. We also conduct extra experiments, demonstrating the effectiveness of our work:
> - On MS-COCO dataset with RetinaNet for object detection, FreeSel still outperforms random selection: 7.0 mAP -> 9.0 mAP (2% data) and 14.3 mAP -> 15.3 mAP (4% data).
> - On ADE20k dataset with Segmenter (ViT backbone) for semantic segmentation, FreeSel outperforms random selection as well: 27.1 mIoU -> 28.2 mIoU (5% data).
>
>
> **Q3: [Select data from web or unannotated data.]**
>
> We agree with the importance of performing data selection algorithms on the web or annotated data. However, as an academic research team, we do not have enough budget to annotate the selected samples to evaluate our method. Besides, most existing active learning algorithms experiment on these standard datasets, so we follow their settings to make a fair comparison. We will consider it as our future work, thanks!
>
> \
> We wish that our response has addressed your concerns, and turns your assessment to the positive side. If you have any questions, please feel free to let us know during the rebuttal window. We appreciate your suggestions and comments!

---

> > ### Comment · Reviewer_yLeE · 2023-08-21
> >
> > I appreciate the authors' response. The rebuttal addressed part of my concerns. Hence, I am keeping my score.

---

> ### Comment · Area_Chair_xpBf · 2023-08-16
> **Discussion?**
>
> Dear Reviewer yLeE,
>
> Any thoughts on authors' detailed response? Further questions / comments on the weakness points you mentioned? Thanks!
>
> Best,
>
> AC

---

> ### Author Response · Authors · 2023-08-21
> **Looking forward to more discussions and your post-rebuttal rating!**
>
> Dear reviewer,
>
> Thanks again for your suggestion to strengthen this work. As the rebuttal period is ending soon, we wonder if our response answers your questions and addresses your concerns. If yes, would you kindly consider raising the score? Thanks again for your very constructive and insightful feedback!
>
> Best, \
> Authors

---

### Official Review · Reviewer_7NQc · 2023-07-05

**Soundness:** 3 good
**Presentation:** 3 good
**Contribution:** 3 good
**Rating:** 6
**Confidence:** 4

**Summary:**

This work introduces a method named FreeSel, which is designed to efficiently select information-rich data points from an unlabeled pool. These selected data points can then be annotated and utilized for training deep learning models, thereby mitigating the costs and resources needed to annotate an entire dataset. The method primarily consists of two key components: feature extraction using a large-scale pretrained model (DINO), and the construction of semantic patterns based on intermediate features. Subsequently, a distance-based selection strategy is employed to identify the most diverse and informative data samples.

The authors assert that the proposed approach addresses three critical aspects: Generality, by decoupling data selection from task-specific models; Efficiency, by enabling the selection of samples in a single pass; and Non-Supervision, by eliminating the need for annotations until the data selection process is complete.

While the results on multiple datasets and tasks indicate that the proposed methodology holds promise, the paper seems to lack some important comparisons. Additionally, characterizing this work as an alternative to traditional active learning may not be entirely accurate.

**Strengths:**

1. The idea of employing a large-scale pre-trained model for data selection is interesting. This is especially pertinent in today's environment, where there is a proliferation of large-scale foundational models. Harnessing these models to select data samples for training of downstream models could prove to be highly advantageous for addressing a wide range of problems.
2. The paper is well-structured and easy to understand
3. The inclusion of experiments across multiple tasks and datasets provides a comprehensive view of the empirical performance of the proposed approach

**Weaknesses:**

1. The primary concern regarding this work pertains to its positioning. The paper seems to be more aligned with the domain of subset selection rather than active learning (AL).
    * AL methods, as the name suggests, actively learn which samples to select in a manner that is specific to the model at hand. While I concur with the authors' perspective that this iterative method is resource-intensive, drawing a direct comparison between the proposed "passive" selection method and traditional active learning in terms of training time may not be entirely appropriate.

2. There is a need for additional baselines to more effectively evaluate the merits of the proposed approach.
    * For instance, in the absence of any iterative active learning, if a random subset of data is selected and a model is trained on it, how does the accuracy of this model compare to one trained on samples selected using FreeSel?

3. There appears to be some conflation between the method (FreeSel) and the problem setting (data selection using large-scale pre-trained models). For example, the principle of Non-supervision (Line 48) seems more pertinent to the methodology than to the problem setting.
4. To gain a clearer understanding of the novelty and contributions of this work, it would be beneficial to compare the method with existing subset selection methods.Such comparisons do not necessarily need to be empirical but could include theoretical or conceptual comparisons.
5. It would be beneficial for the authors to clarify the positioning of FreeSel in relation to traditional active learning and to provide additional comparisons to strengthen the evaluation.

References for consideration:

1. [Kaushal et al., Learning From Less Data: Diversified Subset Selection and Active  Learning in Image Classification Tasks](https://arxiv.org/pdf/1805.11191)
2. [Chang et al., On Training Instance Selection for Few-Shot Neural Text Generation](https://arxiv.org/pdf/2107.03176)
3. [Birodkar et al., Semantic Redundancies in Image-Classification Datasets: The 10% You  Don't Need](https://arxiv.org/pdf/1901.11409)
4. [Ramalingam, et al., Less is more: Selecting informative and diverse subsets with balancing  constraints](https://arxiv.org/pdf/2104.12835)

**Questions:**

1. What is the rationale for choosing CoreSet in Figure, there are much more recent and useful active learning strategies that are specifically proposed for the problem of object detection?
2. No comparisons were made with the works listed in `Data Selection with Pretrained Models` paragraph of related work though they seem very relevant, is there any specific reason for this? Including comparisons with these pertinent works could enhance the comprehensiveness and depth of the evaluation.

**Limitations:**

All the limitations are clearly discussed

---

> ### Author Rebuttal · Authors · 2023-08-09
>
> Thank you for the positive comments and insightful suggestions. The concerns are answered as follows.
>
> **Q1: [Our method is more like subset selection instead of active learning.]**
>
> + We totally agree with your points. Our method should be considered as subset selection more than active learning. That is actually why we name this method "free selection" instead of "XXX active learning". We compare with active learning methods instead of subset selection methods because active learning is the mainstream kind of data selection algorithms.
> + For subset selection algorithms, although they do not require the repetitive batch selection strategy, most of them still need to train a model on a labeled seed subset of the target dataset. Therefore, our method is still much more efficient than theirs since we do not need any extra training on the target dataset. We will make more discussion about the positioning in the final version.
>
>
> **Q2: [Additional baselines for subset selection.]**
>
> In our Fig. 5,6,7, we also provide the random selection baseline (denoted in blue) under different sampling ratios. In the following table, We also give more heuristic subset selection baselines without extra training for SSD object detection model on PASCAL VOC. 1) Perform K-Means on image features extracted by PCA. 2) Perform K-Means on image features extracted by DINO pretrained DeiT. 3）For image features extracted by DINO pretrained DeiT, we calculate the average pairwise feature distances among 10 augmented copies to find hard samples. 4) Using ImageNet supervised pretrained model, we calculate the entropy of prediction to find uncertain samples.
>
>
> | Methods | Num. 3k | Num. 5k | Num. 7k |
> | -------- | -------- | -------- |  -------- |
> | PCA+Kmeans     | 63.87     | 67.77     |  71.02 |
> | DINO+Kmeans     | 64.85     | 68.05     |  71.50 |
> | Inconsistency     | 63.29    | 67.65    |  71.35 |
> | Entropy     | 56.33    | 66.03    |  69.72 |
> | FreeSel     | **65.66**    | **69.24**    |  **71.79** |
>
>
> **Q3: [Some conflation between the method (FreeSel) and the problem setting.]**
>
> Thanks for pointing out. Our problem setting is to select samples with the pretrained model. In this process, we select all the samples in a single run, so we do not have any knowledge about the label space in this process. As a result, this process is naturally unsupervised whatever the detailed selection algorithm is. Therefore, we consider non-supervision as the problem setting instead of the method. We will make more clarification between the problem setting and concrete method in the final version.
>
>
> **Q4: [Compare the method with existing subset selection methods.]**
>
> Thanks for giving those references. They are different from our paper in both problem settings and concrete algorithms. [R1] relies on a small initial seed set to train the model for subset selection, and their subset selection is based on uncertainty instead of diversity. [R2] focuses on the few-shot learning and finetuning of the pretrained model. Their algorithm is based on K-Means, but our answer in Q2 shows that K-Means does not work in our setting. [R3] also trains the model on the target dataset first for the further analysis. Their algorithm is based on the Agglomerative Clustering of global features. Our ablation studies also show that global features cannot solve our problem. [R4] also needs a small initial seed set for model training. Their selection algorithm is a combination of uncertainty and diversity. In contrast to all of them, our method does not need training on the target dataset or knowledge about the label space. We will include more discussion of these methods in the final version.
>
>
> **Q5: [Relation to traditional active learning and additional comparisons.]**
>
> Thanks for your suggestion. We include more explanation of the positioning of our work and make some comparisons with existing subset selection methods in Q1 and Q4. We will also add more information in the final version.
>
>
> **Q6: [Rationale for choosing CoreSet for analysis.]**
>
> As you mentioned in Q1, our problem setting is different from active learning. Thus, we have to adapt those active learning methods to our problem setting for analysis. CoreSet is fully based on the feature diversity and does not require the knowledge of label space, so it can be trivially transplanted to our problem setting while some other methods such as LearnLoss and CDAL cannot be easily adapted. Alternatively, in Fig. 8, we also give results of trivially combining the pretrained model with active learning methods, which does not improve the performance.
>
>
> **Q7: [No comparisons with the works listed in `Data Selection with Pretrained Models` paragraph.]**
>
> + Most papers mentioned in this paragraph are working on other problems different from ours. [28] focuses on the extremely low budget (less than 1%). [39] applies to semi-supervised learning. [42] is targeted for model finetuning with low budgets (mostly less than 5%). Thus, it is hard to compare our method with them directly.
> + Among them, only [44] works on general active learning following the traditional batch selection pipeline. They do not use pretrained models but just borrow some ideas of unsupervised learning for data selection. Their performance is extremely sensitive to the selection of unsupervised learning algorithms, and different unsupervised learning fits different tasks. On CIFAR10, FreeSel performs competitively with PT4AL (rotation) and outperforms PTAL (jigsaw, colorization, SimSiam). However, for segmentation, PT4AL (rotation) performs worst. Therefore, [44] cannot find a general unsupervised learning algorithm that can be used for active learning for all tasks.
>
> \
> Thanks again for your time and effort! For any other questions, please feel free to let us know during the rebuttal window.

---

> > ### Comment · Reviewer_7NQc · 2023-08-19
> >
> > I have carefully reviewed the initial submission, the authors' response, and the feedback from other reviewers. I appreciate the effort that has been invested in addressing the concerns raised, and I would like to thank the authors for the comparison with subset selection methods. The responses provide answerers to my questions, and in light of this, I have updated my rating accordingly.
> >
> > However, I would like to kindly request that the authors further refine the paper by including additional comparisons with relevant approaches. This will not only enhance the robustness of the paper but also provide readers with a clearer understanding of how the proposed method stands relative to existing solutions. Additionally, I encourage the authors to more explicitly position their work within the broader context of the field, either in the final version of this paper or in a future submission. Depending on this positioning, new baselines and comparisons may be necessary to strengthen the paper’s contributions and its distinction from other works in the field.

---

> > > ### Author Response · Authors · 2023-08-20
> > > **Thanks for your response**
> > >
> > > Dear Reviewer 7NQc,
> > >
> > > We sincerely thank you for your reply and for updating the rating. We also appreciate your valuable feedback about our paper. In the camera-ready version, we will follow your suggestions to further improve the paper writing and include additional comparisons.
> > >
> > > Best,
> > >
> > > Authors

---

> ### Comment · Area_Chair_xpBf · 2023-08-16
>
> Dear Reviewer 7NQc,
>
> You gave a initial borderline rating. Any thoughts on authors' response? Further questions / comments? Thanks!
>
> Best,
>
> AC

---

### Official Review · Reviewer_SbPr · 2023-07-06

**Soundness:** 3 good
**Presentation:** 3 good
**Contribution:** 3 good
**Rating:** 6
**Confidence:** 5

**Summary:**

This paper aims to introduce a unique method called FreeSel for data selection. This method is accompanied by a newly designed pipeline that incorporates existing general-purpose models, requiring minimal additional time. By leveraging a publicly available pre-trained model, FreeSel can select data from diverse datasets through a single-pass inference, eliminating the need for extra training or supervision. Specifically, the method involves extracting semantic patterns from intermediate features of the general-purpose model to capture nuanced local information within each image. Furthermore, it enables the selection of all data samples in a single pass using distance-based sampling at a fine-grained semantic pattern level.

**Strengths:**

1- For the first time, this paper presents a novel model that introduces a data selection pipeline. This pipeline incorporates crucial principles of a robust feature extractor, including generality, efficiency, and non-supervision, while incurring minimal time costs. This innovative idea holds potential for application in active learning scenarios.

2- This approach leverages unsupervised extraction of intermediate features by harnessing the benefits of publicly available pre-trained vision transformer (LLM) models. These features are then utilized to define semantic patterns.

3- In contrast to existing active learning approaches, this method selects data examples for a given budget in a single pass, focusing on the diversity found in the local patterns extracted by the pre-trained model. This approach is considerably more time-effective compared to existing methods, as it avoids the need for multiple passes or iterations.

4- To demonstrate the generalizability of the proposed approach, experiments were conducted across various types of tasks, including image classification, object detection, and semantic segmentation. The results of these experiments reveal significant performance improvements, underscoring the effectiveness of the approach across different domains and applications.

**Weaknesses:**

1- We have noticed a decrease in the performance of the proposed model as the number of samples increases, which is in contrast to several active learning approaches. We require a justification for this observation. Is the model sensitive to the size of the datasets?

2- In the paper, the variable "c" represents the number of pseudo categories, which defines the number of visual parts in an image. However, it is unclear how "c" is determined. Is it a fixed value for every image, or does it vary from image to image?

3- The proposed method relies solely on local features to identify semantic patterns, disregarding global features which can be crucial in certain cases. Completely disregarding global features may not be the most desirable approach.

**Questions:**

Please answer all the concerns raised in the weaknesses section.

**Limitations:**

The authors have mentioned the limitations of the proposed approach in the conclusion section. The authors have also presented an ablation study to discriminate the contribution of pre-trained vision transformers (LLM) and other components. It is essential to understand that the performance of the proposed model is not only due to the capability of pre-trained LLM models.

---

> ### Author Rebuttal · Authors · 2023-08-09
>
>
> Thank you for the positive comments and insightful suggestions. The concerns are answered as follows.
>
> **Q1: [Performance decreases as the sample number increases. Sensitivity to the dataset size.]**
>
> - Existing active learning methods train their models on the selected samples after each batch selection step. As a result, as the sample number increases, their models have been trained on a large subset of the target dataset, so it should naturally fit the target domain well. In contrast, our method makes use of a general pretrained without any domain-specific knowledge regarding the target dataset. It is understandable that existing methods can fit the domain of the targeted dataset better after seeing many training samples.
> - Our method is not sensitive to the dataset size.
>     - We conduct experiments on two extra classification datasets with different sizes: CalTech 256 dataset (~30k), CIFAR100 dataset (50k). We do not experiment on ImageNet since it is our pretraining dataset.
>         - For CalTech dataset, we can improve the random selection from 70.2 to 71.4 (15% data) and from 73.7 to 75.8 (25% data).
>         - For CIFAR100, it improves the random selection fron 49.8 to 56.3 (10k samples) and 62.3 to 67.8 (20k samples).
>     - Our experiments in the paper also include PASCAL VOC (16k) and Cityscapes (3k).
>     - These results show that our method is not sensitive to the dataset size.
>
>
> **Q2: [How to determine the visual parts number in each image.]**
>
> We currently set a fixed number for each image. There is an ablation study about this number in the **supplementary materials Tab. 2**, where we find a moderate visual parts number can lead to the best performance.
>
>
> **Q3: [The method disregards the global features.]**
>
> Our method pays more attention to the local features instead of global features since global features are hard to represent complex images (the 1st and 2nd row of Tab. 2 in our paper). However, our method does not totally disregard the global features. Since the pretraining task is image-level contrastive learning, the patch-token features already implicitly encode some useful global information. Besides, we also filter the regions based on the [CLS] token attention map, so the regions unimportant to the image would be discarded.
>
> \
> Thanks again for your time and effort! For any other questions, please feel free to let us know during the rebuttal window.

---

> ### Comment · Area_Chair_xpBf · 2023-08-16
>
> Dear Reviewer SbPr,
>
> Any thoughts on authors' response? Further questions / comments? Thanks!
>
> Best,
> AC

---

> ### Author Response · Authors · 2023-08-21
> **Looking forward to more discussions and your post-rebuttal rating!**
>
> Dear reviewer,
>
> Thanks again for your suggestion to strengthen this work. As the rebuttal period is ending soon, we wonder if our response answers your questions and addresses your concerns. If yes, would you kindly consider raising the score? Thanks again for your very constructive and insightful feedback!
>
> Best, \
> Authors

---

### Official Review · Reviewer_xhjx · 2023-07-14

**Soundness:** 2 fair
**Presentation:** 2 fair
**Contribution:** 2 fair
**Rating:** 3
**Confidence:** 5

**Summary:**

This paper proposed a diversity-based method for active learning. Given an image, features are extracted from a pretrained visual transformer model (DeiT-S), features with [CLS] token attention above a threshold are kept and clustered into K clusters. Each cluster is represented by the average of features belonging to the cluster. The cluster center, named semantic pattern in the paper, is used in a distance-based probabilistic sampling framework (similar to kmeans++) for sample selection. The proposed method is evaluated in various visual tasks including object detection, semantic segmentation and image classification and obtained promising results.

**Strengths:**

1. The proposed method utilized pretrained model for feature extraction and sample selection, eliminating the need to train a model at each AL cycle and thus is much more time efficient than traditional AL methods.
2. The method is evaluated on various tasks and datasets and shown promising results.

**Weaknesses:**

1. The effectiveness of the proposed method is questionable.
(1) From Fig.5 and Fig.7, it can be seen that the proposed method is always outperformed by competing methods at later stage, suggesting  that the pretrained features become less effective in selecting informative samples as the training progresses. Performance at later stage is more important as active learning is only useful when your method can achieve comparable performance (e.g., 95%) of the fully-supervised baseline.
(2) It is not clear how the method performs when the target domain has large domain shift to the pretrained domain. Currently the evaluation datasets are all natural images that are similar to ImageNet. I would expect that the effectiveness of the proposed method would be undermined when the domain shift becomes large.
(3) Benchmarking is not comprehensive, missing BADGE [1] for image classification, EnmsDivproto [2] for object detection.

2. Technical contribution is limited. The proposed method combines existing techniques in a straightforward manner, lacking novelty and technical contribution.


[1] Ash, Jordan T., et al. "Deep batch active learning by diverse, uncertain gradient lower bounds." ICLR2020.
[2] Wu, Jiaxi, Jiaxin Chen, and Di Huang. "Entropy-based active learning for object detection with progressive diversity constraint." CVPR2022.

**Questions:**

The proposed method is essentially doing class-wise aggregation to generate feature representation for an image, which has been explored in CDAL. Apart from the obvious difference that CDAL can obtain pseudo label directly from the target classifier while this paper need to employ clustering to generate pseudo labels, what are other differences and advantages of the proposed method compared to CDAL?

**Limitations:**

The authors discuss the limitation of the proposed method (i.e., cannot beat all state-of-the-art methods) in Section 6 and leave it for future work.

---

> ### Author Rebuttal · Authors · 2023-08-09
>
>
> Thanks for your efforts to improve the quality of our papers. We thank you for pointing out your concerns, which we believe have been carefully clarified and addressed, as follows.
>
> **Q1: [Outperformed by some active learning methods in late stage.]**
>
> Good question! This observation is reasonable, but our method still has wide applicability, such as scenarios where resources are severely constrained.
> - Existing active learning methods train their models on the selected samples after each batch selection step. As the sample number increases, their models have been trained on a large subset of the target dataset, so it should naturally fit the target domain well. In contrast, our method utilizes a general pretrained without any domain-specific knowledge regarding the target dataset. It is understandable that existing methods can fit the domain of the targeted dataset better after seeing many training samples in a later stage.
> - We respectfully push the comment back that data selection is only useful in the late stage, as in the following points.
>     - The budgets in different cases vary significantly. Sometimes the budget can only handle very low sampling ratios such as in medical imaging where labeling requires human experts.
>     - Data is usually cheap and abundant. For example, a vehicle can capture hundreds of images every minute, but we only need to keep a very small part. An efficient data selection algorithm to sample a small percentage of data is really desirable.
>     - Our method can be complementary to SOTA techniques. Since we aim to get a free data selection without any extra training efforts, our method can be easily combined with other techniques like replacing the initial random seed sets for other active learning or subset selection methods.
>
> **Q2: [Performance under large domain shift to the pretrained domain.]**
>
> - We conduct an experiment on RealPizza dataset (multi-label classification with mAP as the metric) with a large domain shift to ImageNet. FreeSel still achieves notable performance gain in comparison to random selection: 10% 35.0->37.5, 15%: 37.8->39.2, 20%: 39.1->40.7, showing its robustness to different domains.
>
> - Besides, we already conduct experiments on datasets from different domains in the paper (PASCAL VOC: natural images, CIFAR: low-resolution images, Cityscapes: driving scenarios). Results show that our method is general enough to most data domains.
>
> **Q3: [Missing comparison with BADGE for classification, EnmsDivproto for detection.]**
>
> + Thanks for pointing out. In Line 213, we explain that we do not compare with active learning methods specifically designed for one task which should naturally perform better than general methods. We will include these related works and analyses in our revision.
>
> + Besides, we assert that performance isn't the sole metric of consideration. We aim to develop a "free" data selection algorithm. We can directly exploit public pretrained model and only need negligible extra efforts for data selection, reducing the data selection time from days to minutes as well as simplifying the data selection pipeline greatly (Line 249-257). In many scenarios, such as selecting data from an extremely large data pool, we believe this advantage of "free" far outweighs minor disparities in performance.
>
> **Q4: [The proposed method combines existing techniques in a straightforward manner, lacking novelty and technical contribution.]**
>
> - Our main novelty lies in the brand-new data selection pipeline. Existing active learning methods need repetitive model training and batch data selection, while other subset selection algorithm also requires to train the model on a small initial subset of the target dataset. The problem is that training models is costly, and the size of subset for training directly affects their performance of data selection. In contrast, our paper designs a novel pipeline to directly exploit general pretrained model to select all the samples from a large data pool in one go. In this case, no training on the target dataset is needed, and data selection time goes down from days to minutes, almost negligible, which means large-scale data selection becomes doable.
> - Although our method is notably straightforward without intricate modules, we believe it's unnecessary to inject excessively complex designs to artificially make it "looks novel". In contrast, it is always desirable that a simple method can finish the task well, especially given that our main novelty lies in the pipeline design. This is verified by our experiment results. If a simple method can rival well-designed existing algorithms in performance, it unquestionably becomes a compelling choice.
>
> **Q5: [Difference with CDAL.]**
>
> We share some common insights into the importance of regional information with CDAL. However, our work is still totally different with CDAL in both pipeline and algorithm.
> 1) The key difference lies in the pipeline. As shown by Reviewer 7NQc, our problem setting and pipeline are quite different with traditional active learning. We exploit public pretrained model to achieve a free data selection with negligible extra efforts. In contrast, CDAL follows the traditional active learning pipeline and requires computationally intensive multi-step training and batch selection.
> 2) Our algorithm is also quite different with CDAL. CDAL relies on a known label space to get the softmax probability for each region, but our algorithm computes the diversity based on the feature clusters which is more friendly to the open set. CDAL aggregates the image-level confusion from region-level softmax probability, but our method directly selects images based on the region-level diversity.
>
> We wish that our response has addressed your concerns, and turns your assessment to the positive side. If you have any questions, please feel free to let us know during the rebuttal window. We appreciate your suggestions and comments!

---

> ### Comment · Area_Chair_xpBf · 2023-08-16
> **Discussion?**
>
> Dear Reviewer xhjx,
>
> Any thoughts on authors' detailed rebuttal? Further questions / comments? Thanks!
>
> Best,
> AC

---

> ### Comment · Reviewer_xhjx · 2023-08-17
>
> Dear authors and AC,
>
> I have read the rebuttal carefully. However, I still don't think the novelty and significance of the proposed method is sufficient for a conference like NeurIPS. Therefore, I will keep my original rating.

---

> > ### Author Response · Authors · 2023-08-17
> > **Definition of novelty and significance**
> >
> > Dear Reviewer xhjx,
> >
> > Thanks for your response. We respect your different opinions from us about the definition of novelty and significance. However, as emphasized in our rebuttal, we believe that novelty is never equivalent to technical complexity. A straightforward algorithm is always compelling as long as it has some desirable properties. Our novelty is mainly reflected by the brand-new pipeline for data selection with several superiorities. We believe a simple free data selection algorithm following a novel pipeline is much more significant for the community than many so-called "novel" and complicated traditional active learning methods.
> >
> > Best,\
> > Authors

---

> > > ### Comment · Reviewer_xhjx · 2023-08-18
> > >
> > > Dear Authors,
> > >
> > > I am not saying that your work has no novelty and significance. I just don't think it is sufficient for a tier-1 conference like NeurIPS: the performance cannot beat state-of-the-art, benchmarking methods are incomplete, method-wise is just some simple combination of existing algorithms. The idea of using pretrained features for active learning is also straightforward.

---

### Author Rebuttal · Authors · 2023-08-09


We sincerely appreciate all reviewers’ time and efforts in reviewing our paper. We are glad to find that reviewers generally recognized our contributions including the novelty of our proposed pipeline (Reviewers SbPr,7NQc), the efficiency of our method (Reviewers xhjx, SbPr, yLeE), detailed experiments on different tasks (Reviewers xhjx, SbPr, 7NQc, yLeE), and our well-written paper (Reviewer 7NQc).


In response to reviewer comments, we have added several new experiments and results to strengthen the paper.

**Summary of new experiments.**
1. New experiments on RealPizza dataset to evaluate the performance under large domain shift. [Reviewer xhjx Q2]
2. New experiments on CalTech256 and CIFAR100 to show the method's ability on datasets of different sizes. [Reviewer SbPR Q1]
3. Extra subset selection baselines. [Reviewer 7NQc Q2]
4. New experiments on ADE20k and MSCOCO to show the performance on modern datasets and networks. [Reviewer yLeE Q2]


We believe these new additions help address reviewer concerns and issues. We thank the reviewers' time and feedback in improving the quality of our work and we hope the revisions further highlight the contributions made. Before answering the point-wise questions raised by reviewers, we first provide the responses to some common concerns, which we believe help reviewers to better position our work and understand our novelty and strengths.


**Response to common concerns.**

**Q1: [Concerns about the novelty and method simplicity.]**

- The main novelty of our work lies in the brand-new data selection pipeline. Existing active learning methods need repetitive model traning and batch data selection, while other subset selection algorithm also requires to train the model on a small initial subset of the target dataset. The problem is that training models is costly, and the size of subset for training directly affects their performance of data selection. In contrast, our paper designs a novel pipeline to directly exploit general pretrained models to select all the samples from a large data pool in one go. In this case, no training on the target dataset is needed, and data selection time goes down from days to minutes, almost negligible, which means large-scale data selection becomes doable.
- Although our method is notably straightforward without intricate modules, we believe it's unnecessary to inject excessively complex designs to artificially make it "looks novel". In contrast, it is always desirable that a simple method can finish the task well, especially given that our main novelty lies in the pipeline design. This is verified by our experiment results. If a simple method can rival well-designed existing algorithms in performance, it unquestionably becomes a compelling choice.



**Q2: [Concerns about the performance and comparison with SOTA active learning methods.]**
- In comparison with SOTA active learning algorithms, we can outperform most of them but underperform some of them in the late stage. Existing active learning methods train their models on the selected samples after each batch selection step. As a result, as the sample number increases, their models have been trained on a large subset of the target dataset, so it should naturally fit the target domain well. In contrast, our method makes use of general pretrained knowledge without any domain-specific training regarding the target dataset. That's why existing methods can fit the domain of the targeted dataset better after seeing many training samples in a later stage. This is reasonable, and our model therefore has wider applicability, such as scenarios where data is too scarce to do batch selection or resources are severely constrained. We believe that the early stage is also critical as training budgets and data vary.
- Nonetheless, we assert that performance isn't the sole metric of consideration. We aim to develop a "free" data selection algorithm. We can directly exploit public pretrained model and only need negligible extra efforts for data selection, reducing the data selection time from days to minutes as well as simplifying the data selection pipeline greatly (Line 249-257). In many scenarios, such as selecting data from an extremely large data pool, we believe this advantage of "free" far outweighs minor disparities in performance.


Please let us know if any clarification or additional experiments would further strengthen the paper. We would be happy to incorporate all these suggestions in the final version. Thank you again for your time and efforts!

---

### Author Response · Authors · 2023-08-16
**Thank you and we are looking forward to your post-rebuttal feedback!**

Dear AC and all reviewers:

Thanks again for all the insightful comments and advice, which helped us improve the paper's quality and clarity.

The discussion phase has been on for several days and we have not heard any post-rebuttal responses yet.

We would love to convince you of the merits of the paper. Please do not hesitate to let us know if there are any additional clarifications that we can offer to make the paper better. We appreciate your comments and advice.

Best, \
Authors

---

### Decision · Program_Chairs · 2023-09-21

**Decision:**

Accept (poster)

**Comment:**

The paper titled "Towards Free Data Selection with General-Purpose Models" proposes a method called FreeSel for data selection in active learning using a pretrained model. The method extracts features from a pretrained visual transformer model, clusters the features into semantic patterns, and uses a distance-based sampling framework for selecting informative samples. The paper highlights the efficiency of FreeSel, showcasing significant time savings compared to traditional active learning methods. The proposed method is evaluated across various visual tasks and datasets.

Reviewer xhjx acknowledges the time-saving advantage of the proposed method. However, the reviewer raises concerns about the effectiveness of the method, pointing out that it is outperformed by competing methods in later stages of active learning. Additionally, the reviewer questions the method's performance under large domain shifts and notes the absence of benchmarking against certain sota techniques. In response, the authors emphasizes on the applicability in scenarios with tight resource constraints. They argue that the effectiveness of data selection is not limited to the late stage and can be valuable for cases where budgets vary significantly or data is abundant. The authors also highlight that their main innovation lies in the pipeline design, enabling large-scale data selection with minimal extra effort. Reviewer xhjx was not convinced and maintains their initial rating as they deem the contribution and significance is not enough.

Reviewer SbPr provides a positive rating (6). They observe a decrease in the method's performance as the number of samples increases, and they inquire whether the model is sensitive to dataset size. They also inquire about the determination of the variable "c," which represents the number of pseudo categories, and its role in the method. Additionally, the reviewer questions the method's disregard for global features and how this approach might affect performance. In response, the authors explain that the performance decrease and demonstrate that their method is not sensitive to dataset size. Regarding the use of local features, the authors mention that their method still considers global features when filtering regions based on attention maps.

Reviewer 7NQc also provide a final positive rating (6). The primary concern raised is regarding the positioning of the paper. The reviewer suggests that the paper aligns more with subset selection than with traditional active learning (AL), and direct comparisons with AL methods might not be suitable. In response, the authors address the concern by including more details about their ablation studies, showcasing the limitations of other subset selection methods in their setting. The authors also clarify the conflation between the method and the problem setting, indicating that the principle of non-supervision applies to the problem setting. Reviewer recommends that the authors further refine the paper by adding more comparisons with relevant approaches. The reviewer also suggests that the authors explicitly position their work within the field's broader context and consider new baselines and comparisons to strengthen the paper's contributions and distinction.

Finally, reviewer yLeE suggest including results on more recent datasets and using modern backbones for evaluation. The reviewer points out that applying FreeSel to select data from web or unannotated sources would provide a stronger demonstration of effectiveness. In response, the authors provide additional experimental results on MS-COCO and ADE20k datasets with different models and backbones. They explain the limitations of performing experiments on web or unannotated data due to resource constraints but express interest in considering this for future work. Reviewers yLeE kept their negative score but did not provide any further justification.

Taking into account the contrasting reviews, the ACs conducted a thorough examination of the paper and concluded that the problem setting itself holds merit, warranting a recommendation for acceptance. However, it is evident that the paper requires significant refinement, particularly in terms of providing a better positioning and a broader context. On a final note, the ACs were reminded of a paper authored by Hou et al titled "Exploring Data-Efficient 3D Scene Understanding with Contrastive Scene Contexts." This work employs a strategy similar to the proposed framework, termed "active labeling," indicating that the proposed general framework could potentially find widespread utility across diverse domains. This observation gains even more significance given the current landscape dominated by large foundational models. In this context, the authors are encouraged to amplify the prominence of this message through improved writing.